# Management and Medical Therapy of Mild Hypercortisolism

**DOI:** 10.3390/ijms222111521

**Published:** 2021-10-26

**Authors:** Vittoria Favero, Arianna Cremaschi, Alberto Falchetti, Agostino Gaudio, Luigi Gennari, Alfredo Scillitani, Fabio Vescini, Valentina Morelli, Carmen Aresta, Iacopo Chiodini

**Affiliations:** 1Department of Medical Biotechnology and Translational Medicine, University of Milan, 20133 Milan, Italy; vittoria.favero@unimi.it (V.F.); arianna.cremaschi@unimi.it (A.C.); iacopo.chiodini@unimi.it (I.C.); 2Department of Endocrine and Metabolic Diseases, IRCCS, Istituto Auxologico Italiano, 20149 Milan, Italy; a.falchetti@auxologico.it; 3Department of Clinical and Experimental Medicine, University of Catania, 95123 Catania, Italy; agostino.gaudio@unict.it; 4Department of Medicine, Surgery and Neurosciences, University of Siena, 53100 Siena, Italy; luigi.gennari@unisi.it; 5Unit of Endocrinology and Diabetology “Casa Sollievo della Sofferenza” Hospital, IRCCS, 71013 San Giovanni Rotondo (FG), Italy; alfredo.scillitani@gmail.com; 6Endocrinology and Metabolism Unit, University-Hospital S. M. Misericordia of Udine, 33100 Udine, Italy; vescini.fabio@aoud.sanita.fvg.it; 7Unit of Endocrinology, Fondazione IRCCS Cà Granda-Ospedale Maggiore Policlinico, 20122 Milan, Italy; valentina.morelli@policlinico.mi.it

**Keywords:** hypercortisolism, adrenal steroidogenesis, glucocorticoid receptor, 11 betahydroxysteroid dehydrogenase, somatostatin, dopamine

## Abstract

Mild hypercortisolism (mHC) is defined as an excessive cortisol secretion, without the classical manifestations of clinically overt Cushing’s syndrome. This condition increases the risk of bone fragility, neuropsychological alterations, hypertension, diabetes, cardiovascular events and mortality. At variance with Cushing’s syndrome, mHC is not rare, with it estimated to be present in up to 2% of individuals older than 60 years, with higher prevalence (up to 10%) in individuals with uncontrolled hypertension and/or diabetes or with unexplainable bone fragility. Measuring cortisol after a 1 mg overnight dexamethasone suppression test is the first-line test for searching for mHC, and the degree of cortisol suppression is associated with the presence of cortisol-related consequences and mortality. Among the additional tests used for diagnosing mHC in doubtful cases, the basal morning plasma adrenocorticotroph hormone, 24-h urinary free cortisol and/or late-night salivary cortisol could be measured, particularly in patients with possible cortisol-related complications, such as hypertension and diabetes. Surgery is considered as a possible therapeutic option in patients with munilateral adrenal incidentalomas and mHC since it improves diabetes and hypertension and reduces the fracture risk. In patients with mHC and bilateral adrenal adenomas, in whom surgery would lead to persistent hypocortisolism, and in patients refusing surgery or in whom surgery is not feasible, medical therapy is needed. Currently, promising though scarce data have been provided on the possible use of pituitary-directed agents, such as the multi-ligand somatostatin analog pasireotide or the dopamine agonist cabergoline for the—nowadays—rare patients with pituitary mHC. In the more frequently adrenal mHC, encouraging data are available for metyrapone, a steroidogenesis inhibitor acting mainly against the adrenal 11-βhydroxylase, while data on osilodrostat and levoketoconazole, other new steroidogenesis inhibitors, are still needed in patients with mHC. Finally, on the basis of promising data with mifepristone, a non-selective glucocorticoid receptor antagonist, in patients with mild cortisol hypersecretion, a randomized placebo-controlled study is ongoing for assessing the efficacy and safety of relacorilant, a selective glucocorticoid receptor antagonist, for patients with mild adrenal hypercortisolism and diabetes mellitus/impaired glucose tolerance and/or uncontrolled systolic hypertension.

## 1. Introduction

Mild hypercortisolism (mHC), also defined as subclinical hypercortisolism, less severe hypercortisolism or subclinical Cushing syndrome (CS), is a condition of excessive cortisol secretion, without the specific symptoms and manifestations of clinically overt CS (i.e., proximal muscle weakness, facial plethora, easy bruising, purple striae) [1,2]. Despite the lack of typical signs and symptoms and the absence of severe cortisol excess, mHC has been associated with bone fragility, mood alterations, hypertension, alterations in glucose and lipid metabolism, increased cardiovascular risk and mortality [3,4,5,6,7,8,9,10,11,12,13]. At variance, patients with mHC may present with features that are common in the general population and less discriminatory for the presence of a cortisol excess, such as a dorso-cervical fat pad (“buffalo hump”), facial fullness, obesity, supraclavicular fullness, thin skin, peripheral edema, acne, hirsutism and poor skin healing. In some patients, mHC may be also completely hidden, and, in this case, it has been called “hidden hypercortisolism” (HidHyCo). Thus, the difference between mHC and HidHyCo is not trivial, as in the former condition, some symptoms may push the patient to seek medical advice, while the latter condition may be found only casually [13]. 

Indeed, a HidHyCo condition is detectable in about 5–30% of patients with adrenal incidentalomas, which are adrenal lesions detected by imaging studies carried out for reasons other than the suspicion of adrenal diseases [14], with a particularly high prevalence in patients with bilateral forms [15,16,17]. As adrenal incidentalomas are thought to be present in up to 7% of individuals above 60 years of age, a condition of HidHyCo is estimated to be present in 0.8–2% of the elderly population, a prevalence, thus, definitely higher than that reported in for CS [18,19]. 

At variance with the numerous data in patients with adrenal incidentalomas, there are scarce data on the frequency of adrenocorticotropin (ACTH)-dependent mHC and, in particular, HidHyCo. In a study by Toini and coauthors on patients with pituitary incidentalomas, the prevalence of HidHyCo has been found to be about 5% [20]. Finally, the overall prevalence of HidHyCo seems to be more prevalent (up to 10%) in high-risk populations such as in diabetic and/or hypertensive patients and individuals with unexplainable bone fragility [13,21]. 

In patients with unilateral adrenal incidentaloma and mHC, the surgical removal of the adrenal mass is suggested by some authors in the presence of consequences of hypercortisolism, since several data show the improvement of T2D and hypertension and the reduction of fracture risk after recovery from cortisol excess [22,23]. In patients with bilateral adrenal incidentalomas and mHC, the complete remission of cortisol hypersecretion can be achieved only performing a bilateral adrenalectomy, which, however, leads to persistent hypocortisolism and a consequent need for lifelong glucocorticoid (GC) replacement therapy [24]. In order to avoid bilateral adrenalectomy, an initial approach with unilateral adrenalectomy has been suggested, aiming to remove the larger and/or more hypersecreting adrenal mass. However, this approach is burdened with a high number of relapses of hypercortisolism [25]. The surgical treatment (i.e., removal of the pituitary adenoma and/or pituitary surgical exploration) in patients with mHC of pituitary origin has never been investigated. In these patients, the treatment of comorbidities possibly related to cortisol excess (i.e., mental health deterioration, osteoporosis, diabetes and hypertension) rather than curing the underlying disease is the only available approach. 

On the basis of these considerations, medical therapy of mHC in patients not applicable for surgery due to the presence of contraindications or the patient’s refusal has been advocated [26]. In this review, we aimed to summarize the available data regarding the medical treatment of mHC, with a particular focus on the molecular pathways that are targeted by the currently available drugs, starting with a brief summary of the criteria for diagnosing mHC. For the sake of simplicity, and since the available studies included both HidHyCo and mHC patients, in the present work we refer only to the latter condition.

## 2. Diagnosis of Mild Hypercortisolism

There is no consensus yet on the clinical and/or biochemical criteria for diagnosing mHC [27], and, of importance, most of the literature focuses on the screening for mHC of adrenal origin only. In the diagnostic algorithm of adrenal masses, many guidelines recommend the use of the 1 mg overnight dexamethasone suppression test (1 mg-DST), due to its better sensitivity in screening for this condition [19,24,28,29,30]. It must be noted that cortisol secretion is a continuum from completely normal to clearly increased levels, and it is highly variable in the same individual. For this reason, evaluating the degree of cortisol secretion should be conducted by considering cortisol levels after 1 mg-DST as a continuous variable, rather than a categorical one [27]. For, at least partially, satisfying this need, other guidelines distinguish different cut-offs of cortisol levels after 1 mg-DST with cortisol levels after 1 mg-DST between 50 and 138 nmol/L (1.9–5.0 μg/dL) or above 138 nmol/L (>5.0 μg/dL) indicating either “possible autonomous cortisol secretion” or “confirmed autonomous cortisol secretion”, respectively [14].

In AI patients, several studies have shown a correlation between cortisol levels after 1 mg-DST and mortality risk, regardless of age and the presence of diabetes and hypertension [6,8,10]. A recent Swedish study not only confirmed these findings, but also showed that the relationship between cortisol levels after 1 mg-DST, mortality and cardiovascular disease in AI patients and mHC is linear up to cortisol levels after 1 mg-DST of 200 nmol/L (7.3 μg/dL) and that a plasma cortisol level of 83 nmol/L (3.0 μg/dL) or higher is associated with a 2- to 3-fold increased mortality risk. If on one hand cortisol after 1 mg-DST, whatever the cut-off, is a highly sensitive diagnostic tool, on the other hand, it is offset by low specificity. Indeed, some authors suggest that if we used cortisol after 1 mg-DST at a cut off of 50 nmol/L (1.8 µg/dL) or of 140 nmol/L (5.1 µg/dL) for diagnosing hypercortisolism, 20% and 5% of the normal population, respectively, would have an abnormal test [9]. 

Reporting all data regarding the different tests used for diagnosing mHC is beyond the scope of the present review. It only seems worth reminding here that additional biochemical tests have been studied [31]. Although none of them were convincing enough for establishing the presence of mHC, they may be used to improve the specificity of the diagnostic approach and to assess the degree of cortisol secretion [32]. Among the additional tests used for diagnosing adrenal mHC, experts agreed on using the measurement of the basal morning plasma adrenocorticotroph hormone (ACTH), 24-h urinary free cortisol and/or late-night salivary cortisol [14], even if, for the latter, the available data are less encouraging [33]. 

Nowadays, more and more authors also agree on the fact that the intention of searching for HidHyCo in patients with increased probability of having mHC, such as those with insulin-treated and/or complicated diabetes, scarcely controlled hypertension and unexplainable bone fragility, highly depends on the sensitivity of the physician rather than on the specificity of the available tests [1,34,35]. Therefore, increasing our awareness in suspecting HidHyCo and our skills in diagnosing mHC will increase the number of patients diagnosed with hypercortisolism, thus requiring appropriate management.

## 3. Management of Mild Hypercortisolism

In overt hypercortisolism of any etiology, surgical resection of the causal tumor is the treatment of choice, while other approaches, such as radiation therapy (for pituitary tumors) or medical therapy, are used when surgery is not possible or not curative [36]. The surgical options include tumor resection in pituitary-dependent hypercortisolism, unilateral adrenalectomy in adrenal hypercortisolism and resection of ACTH-secreting neuroendocrine tumors in hypercortisolism of ectopic source [37]. Bilateral adrenalectomy may be performed in patients with macronodular and micronodular adrenal hyperplasia or in patients with ACTH-dependent Cushing syndrome when a surgical cure is not feasible [37]. Monolateral adrenalectomy in patients with bilateral adrenal hyperplasia is burdened by a high rate of recurrence of hypercortisolism [38].

Besides the biochemical tests, the presence of possible cortisol-related comorbidities are elements to consider for addressing the treatment of choice in patients with mHC. In patients with adrenal incidentalomas and possible mHC, combining the use of cortisol after 1 mg-DST with additional tests and with the clinical features (i.e., presence of hypertension, obesity and diabetes) seems to reliably predict both the clinical amelioration after surgery and clinical worsening if the mHC is left untreated [39,40]. Likewise, the incidence of vertebral fractures is dramatically reduced after adrenalectomy in patients with adrenal incidentalomas and mHC, while it reaches 50% if patients are conservatively managed [23]. 

Although surgery remains the preferable approach for patients with a unilateral adrenal functioning tumor, nowadays and in particular during this period of the COVID-19 pandemic, more and more patients are left untreated due to the scarce availability of hospital beds. Moreover, to date, many patients are diagnosed with bilateral adrenal lesions and in many patients, surgery is not feasible due to advanced age and the high risk of surgical complications. In this setting, medical therapy of mHC is advisable. Notwithstanding this urgent need, data on the efficacy and safety of the classical drugs used to cure mHC are still lacking [5,24]. In patients with adrenal hypercortisolism, nowadays, medical therapy includes adrenal steroidogenesis-inhibiting agents (ketoconazole, metyrapone, mitotane, etomidate) and a GC receptor (GR) blocker (mifepristone). A new steroidogenesis inhibitor has been developed and approved (osilodrostat) and another GR antagonist (relacorilant) is under clinical trial. Finally, very recently in a small study, the 11β-hydroxysteroid dehydrogenase (11BHSD) type 1 (11BHSD1) inhibitor S-707106 demonstrated an effective insulin sensitizer, anti-sarcopenic and anti-obesity effect in both CS and mHC patients [41].

The use of other adrenal blocker drugs, such as etomidate and mitotane, will not be discussed in this review, as they are considered unlikely to have a role in the treatment of mild forms of hypercortisolism. 

The mechanism of action, dose range, adverse effect and clinical effects of the currently available drugs or possible new agents under phase III clinical trials for treating mHC are summarized and figuratively depicted in Figure 1 and Table 1, respectively.

## 4. Medical Therapy of Mild Hypercortisolism

### 4.1. Pituitary-Directed Drugs

To date, very scarce data are available regarding mHC of pituitary origin. Centrally acting agents act on tumorous corticotroph cells to decrease ACTH secretion [42]. The currently available pituitary-directed drugs include the somatostatin (SST) analogue pasireotide and the dopamine (DA) agonist cabergoline (the latter is currently off-label for patients with Cushing’s disease) [43]. The effectiveness of pasireotide and cabergoline is due to the presence of somatostatin receptors (SSTR), including SSTR type 2 and SSTR type 5, and DA receptors, respectively, in most tumorous corticotroph cells [44]. 

#### 4.1.1. Pasireotide

Being localized in the central nervous system interneurons, including hypothalamic ones, SST acts as a neurotransmitter and controls growth hormone secretion [45]. A possible role of SST in modulating pituitary ACTH secretion has also been demonstrated, with SSTR type 2 and SSTR type 5 being the putative target receptors, expressed on the pituitary corticotroph cells [46]. However, in both healthy subjects and patients with pituitary hypercortisolism, ACTH levels do not decrease following the administration of octreotide, an SST analog with high affinity for SSTR type 2, probably due to the SSTR type 2 downregulation caused by the GC excess exposure itself [46]. At variance with SSTR type 2, SSTR type 5 expression is not influenced by GC exposure as shown in cultures pre-incubated with dexamethasone [47] and in rats [48]. On the basis of these data, SSTR type 5 has been proposed as the real target for SST analogs for the therapy of pituitary hypercortisolism, and a multi-ligand SST analog, pasireotide, with a high affinity for SSTR type 5 (158-fold higher than octreotide) has been developed [49]. Pasireotide is formulated as a subcutaneous injection to be administered twice a day (0.3–0.9 mg). In order to increase compliance, pasireotide LAR has been introduced, which represents a long-acting formulation of pasireotide, requiring a single, intramuscular administration every 4 weeks. Both pasireotide and pasireotide LAR have been shown to be effective in controlling hypercortisolism in pituitary hypercortisolism and are approved by the Food and Drug Administration (FDA) and European Medical Agency (EMA) for patients with pituitary hypercortisolism that are not candidates for surgery or in whom surgery has failed [44]. 

The main clinical trial on pasireotide treatment is a double-blind, randomized, multicenter clinical trial involving 162 patients. In this study, about 50% of patients showed substantial amelioration (either ≥50% reduction from baseline or normalization) in UFC levels at six months. A significant amelioration even of the lipid profile and blood pressure and a reduction in body mass index (BMI) and waist circumference were found in the vast majority of patients, partially independently of full disease-activity control. Importantly, the greatest blood pressure decrease was observed in patients who did not receive antihypertensive medications during the study, and normalization of UFC levels was observed more frequently in patients with mHC, defined as a UFC level between 1.5 and 2 times the upper limit of the normal range (ULN) [43,50]. A subsequent post-marketing observational study reported an even higher success rate (67.7%) in patients with very mild (UFC < 1.5* ULN) to moderate disease (>2 ≤ 5* ULN), confirming that the lower the level of UFC (and therefore the severity of the disease), the higher the rate of UFC normalization [51]. It should be noted that, in this study, three patients with confirmed Cushing’s disease but with normal UFC levels at baseline were treated with pasireotide, obtaining marked beneficial effects in the overall clinical picture (mainly a reduction in BMI, waist circumference and blood pressure) [52]. Pasireotide LAR was also shown to be effective in a phase III study on 150 hypercortisolemic patients. Even in this study, approximately 40% of patients showed normalization of UFC, and higher response rates (52%) were documented in patients with lower baseline UFC concentrations (<2* ULN). The clinical profile after 12 months of treatment was similar to that of twice-daily subcutaneous pasireotide [52]. Pasireotide and pasireotide LAR adverse effects include abdominal discomfort, bloating, diarrhea, cholelithiasis, and bradycardia, and, of importance, hyperglycemia, which appears as soon as 15 days after the introduction of the drug [53]. This side-effect is independent of the hypercortisolism control and is a consequence of insulin secretion inhibition/suppression of incretin release from the gut [54] and, therefore, careful monitoring of hyperglycemia is mandatory in patients treated with these agents. Given this evidence, the use of pasireotide and pasireotide LAR could be considered in patients with pituitary mHC in the absence of diabetes.

#### 4.1.2. Cabergoline

Cabergoline is a DA agonist, acting on the DA receptors with the highest affinity for DA receptors type 2 (D2R), which are expressed in approximately 60–75% of corticotroph adenomas [51]. Preclinical data on mice and cellular studies suggested D2R as a potential target for treating patients with pituitary cortisol excess. Indeed, in rodents, the hypothalamic DA neurons, acting through D2R, maintain the intermediate pituitary lobe under tonic inhibition [55]. Similarly, in D2R-deficient mice, a hypertrophic intermediate pituitary lobe and increased ACTH and proopiomelanocortin (the ACTH precursor) expression has been described. In addition, bromocriptine, a DA agonist, has been shown to inhibit the proliferation of murine ACTH-secreting pituitary adenoma cells [56] and cabergoline treatment in primary cultures of canine ACTH-secreting adenomas moderately expressing D2R-reduced ACTH secretion [57]. It is still unclear whether, in humans, the pituitary ACTH characteristics may influence the D2R expression, even though the D2R expression is thought to be present across all corticotroph subtypes [58]. Interestingly, in ACTH-secreting pituitary adenomas causing mHC, D2R expression has been found to be lower than in silent ACTH-producing adenomas [59]. Thus, it is not possible to exclude that the tumor features may influence D2R expression levels, potentially affecting DA agonists’ effectiveness [60], particularly in mild pituitary hypercortisolism.

Cabergoline is currently suggested as a second-line, off-label treatment for pituitary hypercortisolism [42,43,44]. A meta-analysis on the use of cabergoline monotherapy in patients with pituitary hypercortisolism showed that the proportion of patients achieving remission was 39.4%. Noteworthily, the meta-analysis showed that responder patients had significantly lower baseline UFC than non-responder patients, suggesting a higher chance of success in patients with mHC as compared to patients with severe disease. However, during a long-term follow-up, 8 out of 36 patients (22.2%) who responded to cabergoline monotherapy experienced treatment escape [61]. The reasons for this escape are still poorly understood. Interestingly, in a study on human-induced pluripotent stem cells, cabergoline was described as an inducer of 3β-hydroxysteroid dehydrogenase, an essential enzyme for adrenogonadal steroidogenesis. Although cabergoline is a potent D2R agonist, it may act, though with lower affinity, even on DA receptor type 1 (D1R), which has been shown to upregulate the expression of various steroidogenic enzymes and increase the secretion of steroid hormones synergistically with ACTH [62]. Thus, cabergoline could exert a bimodal effect in patients with pituitary mHC, as it may inhibit ACTH secretion by acting on D2R at the pituitary level, and induce steroidogenesis by acting on D1R at the adrenal level. Moreover, from a clinical perspective, cabergoline treatment seems to also have beneficial effects in cortisol-related comorbidities, mainly arterial hypertension and glycemic control and body weight [63].

Cabergoline therapy is usually well-tolerated, even though some patients actually report nausea, vomiting and orthostatic dizziness. In a lower percentage of patients, headache, nasal congestion, constipation, digital vasospasm, nightmares, anxiety and depression may occur [43]. In general, cabergoline may represent an effective treatment in patients with pituitary mHC when considering its low cost, the oral route of administration and good tolerance.

### 4.2. Adrenal Steroidogenesis Inhibiting Agents

#### 4.2.1. Metyrapone

Metyrapone is a steroidogenesis inhibitor acting mainly against the adrenal 11β-hydroxylase, a crucial enzyme responsible for the conversion of 11-deoxycortisol in cortisol and of 11-deoxycorticosterone in corticosterone, an aldosterone precursor. Metyrapone therefore induces a decrease in GC and mineralocorticoid production and secretion. Moreover, it is also an inhibitor, to a lesser degree, of 18-hydroxylase [64]. Metyrapone was also demonstrated to have an extra-adrenal effect, as it influences the peripheral GC metabolism through the regulation of 11BHSD1 activity [65]. The metyrapone immediate-release capsule, which contains a 250 mg dose with a maximum daily dose of 6 g/day, is suggested [42,66]. Metyrapone is administered orally and absorbed quickly. With a short half-life (2 h) and a peak plasma concentration within 1 h from ingestion, this agent represents a highly manageable treatment option in patients with hypercortisolism. Metyrapone, according to a recent meta-analysis, has an estimated average CS remission rate of 75.9% [67]. Metyrapone treatment substantially reduces serum cortisol and aldosterone, while it increases the androgenic and mineralocorticoid precursors (predominantly 11-deoxycorticosterone, 11-deoxycortisol) levels. About 50–70% of CS patients treated with metyrapone experience the normalization of cortisol parameters. It is essential that any cortisol assay used for monitoring patients treated with metyrapone does not cross react with 11-deoxyxcortisol to avoid an erroneous increase in drug dose due to an apparent increase in plasma cortisol levels [68].

Metyrapone has been reported to improve hypertension, glucose metabolism and psychiatric disturbances in CS patients. In a single-center retrospective study on 91 patients, psychiatric disturbances, glucose intolerance or diabetes mellitus and hypertension improved in 73%, 82% and 70% of patients, respectively [69]. As reported by an observational study on 31 patients, the effect of metyrapone on blood pressure is variable, with the new onset of hypertension or its worsening being described in about 28.6% and 20.8% of cases, respectively, while blood pressure amelioration or normalization occurred in 16.6% of patients [70]. To date, there is still little evidence for the use of metyrapone in patients with mHC. In a recent proof-of-concept study, metyrapone was administered in six patients with autonomous cortisol secretion and adrenal incidentaloma. In these patients with proved increased evening and nocturnal cortisol levels, metyrapone was administered with the aim of restoring the normal cortisol circadian rhythm. In detail, a first metyrapone dose (500 mg) was administered at 6 PM and a second dose (250 mg) was administered at 10 pm. This administration schedule led to a correction of the abnormal circadian rhythm of cortisol secretion, thus suggesting the possible use of low doses of metyrapone in patients with mHC with the aim to restore the physiological rhythm of cortisol [71]. 

The increased levels of androgenic precursors may lead to hirsutism and acne, while increasing levels of the mineralocorticoid precursor can lead to edema, hypokalemia and hypertension at high doses of the drug [64]. Moreover, the loss of negative feedback due to low circulating cortisol levels may then lead to an increase in ACTH levels, which drives the accumulation of cortisol and aldosterone precursors and androgens in blood and urine [70]. The most frequent side effects of metyrapone are gastrointestinal disturbances such as nausea and abdominal discomfort, with other side effects such as hirsutism in females, dizziness and arthralgias being less common [64]. The rationale of using metyrapone in subjects affected by ACTH-independent hypercortisolism is that, in these patients, the compensatory rise of ACTH secretion during treatment should not be observed, since ACTH is chronically suppressed by the autonomous cortisol secretion. As a consequence, in these patients, the side effects caused by an ACTH-mediated increase in steroid precursors with weak mineralocorticoid activity (increased blood pressure, edema and hypokalemia) or androgen effects (hirsutism, acne, irregular menstrual cycles) should not occur. This has been confirmed in the only available prospective study on seven patients, in whom a short-term (i.e., 3 months) metyrapone course at a mean dose of 750 mg/day was given as preoperative therapy in patients with adrenal hypercortisolism [72]. Indeed, in this study, all patients showed normalization of UFC levels from baseline to the end of the follow-up with a reduction of serum and salivary cortisol levels, and no significant increase in plasma ACTH and serum dehydroepiandrosterone sulfate levels [72]. The quality of life and blood pressure control were ameliorated in all patients, while no significant change in weight, electrolytes, or glycemic and lipid profile was reported. In spite of a significant increase in testosterone and androstenedione levels in women, clinical hyperandrogenism did not worsen and, in general, all drug-related adverse events were grade 1 or 2 and generally transient. Of note, three out of seven patients were affected by mHC (i.e., UFC levels less than 1.5-fold higher than the upper limit of the normal range) [72]. 

Finally, very recently, a Japanese research group described the case of a patient affected by pituitary hypercortisolism, who showed GC-driven positive feedback and both ACTH suppression and tumor shrinkage by metyrapone [73]. The same authors revealed that in a cohort study, 8.7% of patients with pituitary hypercortisolism may display a GC-driven positive-feedback, representing a possible new subtype of pituitary hypercortisolism. The molecular mechanisms underlying these effects of metyrapone are still unknown, but it may be due to a GC-mediated effect on the corticotrophs. Indeed, it has been shown that the GC dexamethasone is able to suppress tumor growth by inhibiting tumor angiogenesis by reducing interleukin-8 and endothelial growth factors and by upregulating microRNA-708 expression. Other data pointed out that dexamethasone may potentially enhance tumor growth by inhibiting p53-mediated apoptosis and stimulating protein kinase B and the mitogen-activated protein kinase [74]. Moreover, recent data on murine corticotroph AtT20 cells showed that dexamethasone causes an increase and a decrease in the expression of β-arrestin 1 and β-arrestin 2, respectively, which are proteins involved in the regulation of SST and DA receptors [75]. However, consistent with a direct effect of metyrapone on ACTH-secreting cells is the finding of spontaneous ACTH normalization and tumor regression induced by metyrapone in a patient with ectopic ACTH syndrome [76].

Overall, these preliminary data suggest that metyrapone could represent an option for medical therapy even in mHC. In Europe, metyrapone has been approved for the management of CS by EMA in April 2014, while it is still an off-label treatment in the United States.

#### 4.2.2. Ketoconazole

Ketoconazole (KTZ) is an imidazole derivative, originally conceived as an orally active antifungal agent. In fungi, KTZ inhibits the synthesis of ergosterol, a cell membrane sterol, through the enzymatic blockade of several fungal steroidogenesis enzymes [77]. However, later in vitro and in vivo studies showed that KTZ was able to inhibit adrenal steroidogenesis [78]. Ketoconazole has a relatively short half-life (3.3 h), therefore requiring a twice- or thrice-daily administration schedule. It is formulated as an immediate-release tablet, containing a 200 mg dose, with a maximum total daily dose of 1200 mg/day [42]. It is currently available as a 50/50 racemic mixture of two enantiomers, levoketoconazole (2S,4R stereoisomer) and dextroketoconazole (2R,4S stereoisomer), with these enantiomers exhibiting differences in inhibitory potency for the enzymes involved in steroidogenesis [79]. In the adrenal cortex, KTZ blocks multiple steps of steroid biosynthesis through the inhibition of cytochrome p450 enzymes 17a-hydroxylase, 20,22-desmolase, 11b-hydroxylase, 17,20-desmolase and 18-hydroxylase, therefore inducing a decrease in GC, mineralocorticoid and adrenal androgen production and secretion [64]. Ketoconazole has also been reported to directly inhibit ACTH secretion, thus potentially suggesting a double pharmacodynamic action of potential use in patients with pituitary hypercortisolism, although these mechanisms are still debated [79]. Finally, as for metyrapone, a possible inhibitory effect of KTZ on corticotrophs has been hypothesized. Indeed, in vitro data found that KTZ inhibited ACTH secretion at therapeutic doses by impairing adenylate cyclase activation in corticotrophs [80].

Noteworthily, KTZ impairs not only adrenal but also gonadal steroidogenesis, in particular with a negative effect on testicular androgen production, thus potentially leading to male hypogonadism [42]. Importantly, ketoconazole is a potent inhibitor of CYP3A4 and thus may increase the availability of drugs metabolized through this enzyme [43]. Ketoconazole is also listed among drugs that prolong the QT interval and may increase the risk of torsade de pointes. This possible adverse event, however, was not confirmed in a study that found that long-term ketoconazole administration does not appear to be associated with a significant prolongation of the QT interval in patients with CD [77]. Moreover, KTZ inhibits liver enzymes involved in the metabolism of chemical substances not normally found or expected to be present in the human organism, therefore favoring the occurrence of liver damage [42]. In a metanalysis, Yan and coauthors showed that the incidence of KTZ-associated hepatotoxicity was 3.6–4.2%, independently of the dosage and treatment duration [81]. In the majority of cases, KTZ caused an asymptomatic increase in liver transaminases. However, fatal hepatitis has been reported in patients receiving 200 mg daily after both short- and long-term treatment courses [64]. This is why the FDA has mandated the insertion of a “black box” warning on the ketoconazole label to inform patients and physicians of the risk of serious hepatotoxicity associated with medication use and recommend regular monitoring of liver chemistries [82]. Overall, the main adverse events (12–15% frequency) associated with KTZ use are liver enzymes elevation, gastrointestinal disturbances, gynecomastia and adrenal insufficiency [42].

One recent metanalysis reported an estimated average remission rate of hypercortisolism with the use of KTZ of about 71% [67]. Although it has been used for decades, no prospective studies are available regarding the effect of KTZ for treating the hypercortisolemic states [25]. A French study retrospectively assessing data from 38 patients with CS receiving ketoconazole for a median of 23 months showed, 3–6 months after the initiation of KTZ, blood pressure normalization in all patients, an improvement in metabolic control in all diabetic patients, a marked regression of cardiac signs in a patient who had severe heart failure and a dramatic increase in bone mineral density in three patients followed up for at least 36 months [83]. The largest retrospective study on the use of KTZ in CS, including 200 patients from 14 centers in France treated for more than 24 months, showed the improvement of hypertension, diabetes and hypokalemia in 55.5%, 50% and 87.5% of patients, respectively [84].

The possible use of KTZ in patients with mHC has been scarcely investigated. Comte-Perret et al. reported a case of a 48-year-old woman with bilateral macronodular adrenal hyperplasia (BMAH) with biochemical hypercortisolism without specific signs of CS, treated for 10 years with low doses (200 to 400 mg/day) of ketoconazole to control cortisol secretion. At diagnosis, the patient presented with marked hypertension, which rapidly normalized and required only small doses of spironolactone and metoprolol after the beginning of KTZ. Ketoconazole therapy caused rapid normalization of cortisol and ACTH that persisted over 10 years on treatment, with no adrenal changes in size [85]. To date, we still do not know if KTZ may also be used in mHC, since the side effects may outweigh the benefits. In November 2014, KTZ was approved for CS treatment by EMA, whereas it is still an off-label treatment in the United States [42].

#### 4.2.3. Levoketoconazole

Levoketoconazole (levoKTZ), an orally administered KTZ stereoisomer, is currently under clinical trials for the treatment of CS. LevoKTZ, formulated as a 150 mg immediate-release tablet, is the purified 2S,4R enantiomer of KTZ. Based on early in vitro analyses, levoKTZ inhibits cytochrome P450 family 11 subfamily B member 1 (CYP11β1, coding for 11β-hydroxylase enzyme), cytochrome P450 Family 17 Subfamily A Member 1 (CYP17α1, coding for 17α-hydroxylase enzyme) and cytochrome P450 family 21 subfamily A member 2 (CYP21α2, coding 21α-hydroxylase enzyme), 15- to 25-fold more potently compared to the 2R,4S KTZ enantiomer. This may allow for a lower dose of levoKTZ compared to racemic KTZ to achieve the same efficacy and, consequently, less hepatotoxic effects. Together with a favorable safety profile and an increased therapeutic index, levoKTZ is a promising novel treatment option for CS [82]. Similar to ketoconazole, levoKTZ is a substrate and a potent inhibitor of a major drug-metabolizing enzyme, CYP3A4, potentially leading to relevant drug–drug interactions [86].

LevoKTZ was initially investigated as a potential treatment for type 2 diabetes mellitus [87]. In a small, randomized, double-blind, placebo-controlled study in patients with type 2 diabetes mellitus without hypercortisolism, levoKTZ ameliorated lipid levels and reduced body weight and blood pressure [88]. However, levoKTZ therapy was associated with gastrointestinal side effects and the development of this drug for type 2 diabetes was interrupted due to safety concerns [64,88,89]. The mechanisms explaining the beneficial effects of levoKTZ on diabetes, lipid profile and hypertension, regardless of the presence of hypercortisolism, are still largely unknown. However, it is known that diabetic patients may have increased hypothalamic–pituitary–adrenal axis activity, in particular in the presence of diabetic complications [90,91]. In addition, recent data show that the presence of hypertension and/or diabetes and/or fragility fractures is associated with cortisol secretion, peripheral activation (as reflected by the 11BHSD1 activity) and GC sensitivity (as mirrored by the GR gene sensitizing variants). Therefore, it is not possible to exclude that modulating cortisol secretion may be beneficial for lipid levels, glycometabolic and blood pressure control even in patients without documented cortisol hypersecretion [92,93].

On the basis of these promising results on glucose metabolism, levoKTZ was studied in 94 patients affected by CS in a phase III, open-label, multicenter clinical trial (SONICS study), which found UFC normalization in 46% and 33% of patients with diabetes mellitus and without diabetes mellitus, respectively. Interestingly, in diabetic patients, who were adequately controlled at baseline, levoKTZ treatment led to a clear amelioration of glycometabolic control (i.e., glycated hemoglobin decreased from 6.9% to 6.2%), and a decrease in glycated hemoglobin (i.e., from 5.5% to 5.3%) was noted even in non-diabetic patients [94]. In addition, after the 6-month maintenance period, a significant decrease in body weight, peripheral edema and acne and hirsutism in women as well as an improvement in quality of life and depressive status was observed [94,95]. These data now need to be confirmed in the phase III, double-blind, multicenter clinical trial (LOGICS study) that was started in September 2017 to assess the efficacy and safety of levoKTZ treatment in CS patients, of which no preliminary data have been available so far (https://clinicaltrials.gov/ct2/show/NCT03277690; accessed date: 21 October 2021).

Similar to what was already shown for KTZ, levoKTZ also seems to inhibit cell growth and ACTH secretion in mouse pituitary tumor cells (AtT20) and in primary human pituitary adenoma culture [96].

Given the beneficial effects of levoKTZ on diabetes mellitus and hypertension, even in patients without documented hypercortisolism, and the possibility to use lower doses as compared with KTZ, levoKTZ seems to be a possible effective treatment for mHC, in particular in patients with HidHyCo found during the screening for possible secondary forms of diabetes mellitus and hypertension.

#### 4.2.4. Osilodrostat

Osilodrostat (LCI699) is a novel oral steroidogenesis inhibitor, licensed by EMA in January 2020 for the treatment of endogenous CS in adults and by the FDA in March 2020 for the treatment of patients with pituitary hypercortisolism, who either cannot undergo pituitary surgery or have persistent disease after pituitary surgery [42]. Osilodrostat is a steroidogenesis inhibitor that inhibits the enzyme CYP11β1, which is responsible for the conversion of 11-deoxycortisol to cortisol and of 11-deoxycorticosterone to corticosterone. The drug also inhibits aldosterone synthase (CYP11β2), the enzyme involved in the conversion of 11-deoxycorticosterone to aldosterone [97]. Therefore, similarly to metyrapone, it induces a decrease in GC and mineralocorticoid production and secretion [97]. Osilodrostat only modestly suppresses androstenedione, dehydroepiandrosterone-sulphate, testosterone and 17-hydroxyprogesterone production, whereas it increases progesterone production [68]. Osilodrostat is formulated as immediate-release tablets, with a maximum dose of 60 mg/day and a 4-h half-life, allowing a twice-daily administration [98]. Osilodrostat clinical efficacy and tolerability have been proven in phase 2 and 3 trials with CD patients who have had an inadequate response to transsphenoidal surgery (TSS) after conventional first-line treatment [98]. A complete biochemical response, occurring as early as 12 weeks, has been documented in 66% of patients at 48 weeks with clinical improvements being observed in most cardiovascular-related metabolic parameters including body weight, BMI, fasting plasma glucose, blood pressure and lipid profile [99]. 

Osilodrostat is generally well tolerated with relatively rare side effects, including nausea, diarrhea, fatigue, headache, oedema and hypokalemia, together with hirsutism and acne in women [98,99].

In patients with mHC, due to the reported possible occurrence of cortisol withdrawal syndrome related to the great potency of osilodrostat, early and frequent clinical monitoring is needed, particularly in the first weeks of treatment. The risk of delayed acute adrenal insufficiency in patients treated with a stable dose of osilodrostat has also been recently documented [100].

### 4.3. Glucocorticoid Receptor-Directed Drugs

#### 4.3.1. Mifepristone

Mifepristone is an oral non-selective GR antagonist, officially approved by EMA in February 2012 for the treatment of CS of any cause, if the patient has glucose intolerance and hypertension, or both, and cannot undergo or refuses surgery [42,68]. In Europe, no official approval has been granted to mifepristone as a treatment for CS [68]. Mifepristone has a rapid onset of action and it is prescribed with a maximum daily dose of 1200 mg. Its long half-life (24–90 h) allows a once-daily administration [68]. Mifepristone was discovered in the early 1980s and due to its anti-progestin activity, it has been used as a contraceptive [101].

Thereafter, mifepristone at higher doses was demonstrated to have an inhibitory effect on GR [102]. Indeed, mifepristone directly acts on GR and, thus, it does not reduce cortisol secretion but rather inhibits its peripheral effects [68]. As a consequence, by blocking the GR, mifepristone raises the secretion of ACTH and, consequently, cortisol levels are increased. If this increase in cortisol levels exceeds the capacity of the renal 11BHSD type 2 (11BHSD2) enzyme to inactivate cortisol, a state of apparent mineralocorticoid excess, including hypokalemia, arterial hypertension and peripheral edema, may occur [46]. Moreover, being a non-selective steroid receptor antagonist, mifepristone also binds androgen and progestin receptors, inhibiting their peripheral effects. In particular, the anti-progestin effect is characterized by endometrial thickening and abnormal vaginal bleeding [103], which, however, occurs rarely. The most common adverse effects of mifepristone include nausea and fatigue, possible manifestations of adrenal insufficiency. As the mifepristone mechanism of GR antagonism does not allow one to rely on the commonly used biochemical parameters of hypercortisolism due to the compensatory rise in circulating cortisol levels, the evaluation of efficacy is exclusively based on clinical symptoms [43].

A study evaluating the clinical effects of mifepristone monotherapy in 50 patients with CS found an improvement in hypertension and glucose tolerance in 38% and 60% of patients, respectively. Overall, an improvement in clinical status was observed in 87% of patients, mirrored by the amelioration of depression, cognitive function and quality of life [104]. Mifepristone ameliorated weight, and this beneficial effect on weight loss persisted for an additional two years in patients who remained on therapy [105]. Mifepristone was also studied in patients with mHC and bilateral macronodular adrenal hyperplasia (BMAH). In their first study in 2013, Debono and colleagues treated six individuals with adrenal incidentalomas and mHC with mifepristone 200 mg twice daily for 4 weeks. Across the group, there was a significant reduction in insulin resistance as five out of six individuals showed a reduction in insulin, and in two patients, a clinically significant cardiovascular benefit was shown [106]. These results were confirmed by the second study performed in eight patients with mHC and unilateral or bilateral adrenal incidentalomas. Mifepristone, administered 300 mg daily over 6 months, led to a significant reduction in insulin resistance as measured and an improvement in Beck’s Depression Inventory scores and Cushing’s Quality of Life scores in most patients [107]. In the same year, other authors showed that mifepristone was able to induce an improvement in cardiometabolic parameters as early as 2 weeks after treatment initiation in 4 patients with BMAH, who also experienced an amelioration of glycemic control and hypertension as well as weight loss [108]. Overall, the possible use of mifepristone as a treatment of adrenal mHC for patients with impaired glucose metabolism is hindered by its adverse events (abnormal vaginal bleeding, edema and potential hypocortisolism).

#### 4.3.2. Relacorilant

Relacorilant is a novel oral GR antagonist, currently under clinical evaluation for CS and mHC treatment [42]. This drug has been developed to overcome the mifepristone limits, related to the low selectivity in antagonizing the GR.

A single-arm, open-label, phase 2, dose-finding study with two dose groups was conducted between June 2016 and September 2018 at 19 sites in the USA and Europe. Low-dose relacorilant (100–200 mg/d) or high-dose relacorilant was administered for 12 weeks in 17 CS patients or for 16 weeks in 18 CS patients, respectively. Doses were up-titrated by 50 mg every 4 weeks. Hypertension was ameliorated in 41.7% and 63.6% of patients and glucose metabolism improved in 15.4% and 50.0% of patients in the low-dose group and high dose group, respectively. The most common adverse events included back pain, headache, peripheral edema, nausea, pain at extremities, diarrhea and dizziness, with no patients reporting vaginal bleeding or hypokalemia [109]. Currently, a phase III, randomized, double-blind, placebo-controlled, clinical trial (GRACE, NCT 03697109) is ongoing with the aim of assessing the efficacy and safety of relacorilant treatment in CS patients.

Given its more selective effect on GR and the better safety profile as compared with mifepristone and the promising results with this latter drug in patients with mHC, relacorilant represents an interesting possibility for patients affected with mHC and hypertension or glucose metabolism impairment. Currently, a phase III, randomized, double-blind, placebo-controlled study (GRADIENT, NCT04308590) is ongoing with the aim of assessing the efficacy and safety of relacorilant to treat hypercortisolism in patients with cortisol-secreting adrenal adenoma or hyperplasia associated with diabetes mellitus/impaired glucose tolerance and/or uncontrolled systolic hypertension.

## 5. Conclusions

Mild hypercortisolism is a relatively common condition that may lead to negative metabolic, cardiovascular and psychological outcomes. To date, no guidelines are available regarding its management. From a pharmacological perspective, only metyrapone and mifepristone have been recently studied in this setting specifically, with interesting results, that do, however, need to be confirmed in larger studies. Pasireotide and cabergoline also seem to have potential use in pituitary mHC, taking care to avoid pasireotide in patients with glucose impairment. Even the use of ketoconazole can be considered in mHC patients, carefully evaluating the risk of potential adverse effects as well as potential drug–drug interactions. Nowadays, these drugs could be considered in the cure of mHC in patients in whom surgery is refused or contra-indicated and in whom hypertension and diabetes control is of concern. Besides levoKTZ and osilodrostat, which could be of interest in future studies, and relacorilant, which is now on trial, in the future, other molecular targets could be considered. Indeed, the 11BHSD1 enzyme is among the determinants of the GC exposure at the peripheral target tissues and its modulation has been already advocated as a possible drug target for treating hypercortisolemic states, both functional and tumor induced [35]. Indeed, selective inhibitors of the hepatic 11BHSD1 have been suggested to be potentially useful for patients with mHC and insulin resistance and/or diabetes [41,110], but also in eucortisolemic patients with diabetes [111].

Given the not-negligible prevalence of mHC and the importance of its consequences, further studies on potential therapeutic targets for mHC are warranted.

## Figures and Tables

**Figure 1 ijms-22-11521-f001:**
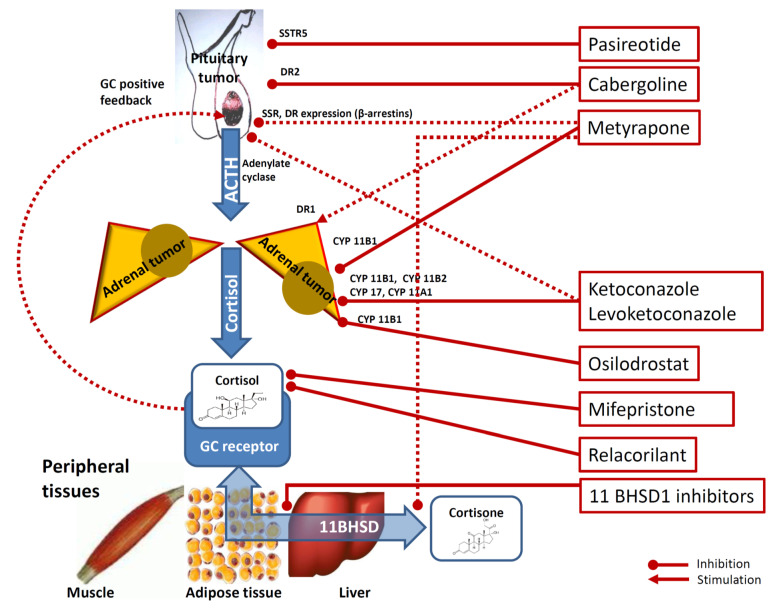
Mechanisms underlying the effects of medical therapy of hypercortisolism. Footnotes: Cabergoline inhibit pituitary corticotrophs via dopamine receptor type 2 (DR2). Cabergoline is also thought to exert a stimulatory role on cortisol secretion on adrenal cortex cells via DR1 dopamine receptor type 2. Pasireotide inhibits pituitary corticotrophs via somatostatine receptors type 5 (SSTR5). Metyrapone inhibits adrenal 11-βhydroxylase and, to a lesser degree, 18-hydroxylase (CYP11B1). Metyrapone is also hypothesized to reduce at the peripheral target tissues (i.e., muscle, adipocytes and liver) the conversion of cortisone into the more active cortisol via modulation of 11beta-hydroxysteroido-dehydrogenase (11BHSD) and to inhibit pituitary corticotrophs via reduction of GC-driven positive-feedback and via SSTR5 and DR2 receptors expression via the modulation of β-arrestin 1 and β-arrestin 2 expression. Ketoconazole and levoketoconazole act on the adrenal steroidogenesis via 11-βhydroxylase (CYP11B1) inhibition, 18-hydroxylase (CYP11B2) inhibition, 20,22-desmolase (CYP11A1) inhibition and on 17a-hydroxylase and 17,20-desmolase (CYP17) inhibition. Ketoconazole and levoketokonazole are also hypothesized to inhibit pituitary corticotroph inhibition by impairing adenylate cyclase activation. Osilodrostat is a steroidogenesis inhibitor acting on CYP11B1) via 11-βhydroxylase inhibition. Mifepristone inhibits the peripheral effects of glucocorticoids (GC) by non-selectively antagonizing the GC receptor. Relacorilant is a selective inhibitor of GC receptor. Finally, some 11 beta-hydroxysteroid dehydrogenase (11BHSD) type 1 (11BHSD1) inhibitors (for example INCB13739, S-707106 and chenodeoxycholic acid) have been suggested to decrease cortisone-to-cortisol conversion, therefore reducing the amount of cortisol (more active) at peripheral target tissues level. Dotted lines are used for not clearly demonstrated pathways.

**Table 1 ijms-22-11521-t001:** Summary of mechanisms of action, dose range, adverse effect and clinical effects of the currently available drugs or possible new agents for treating mild hypercortisolism.

Drug Name	Mechanism of Action	Usual Dose Range in CS and Indication	Possible Schedule in mHC	Main Adverse Events in CS Patients (%)	Data on Possible Use in mHC
Pasireotide	SS receptor agonist - corticotrophs inhibition	- 0.3–0.9 mg SC bid - CD	Never tested	- Diarrhea (45–70) - Hyperglycemia (70–90) - Nausea (25–70)	No data available in mHC Hypothetical use in pituitary mHC in the absence of diabetes
Pasireotide LAR	SS receptor agonist - corticotrophs inhibition	- 10–30 mg IM every 4 wk - CD	Never tested	- Diarrhea (45–70) - Hyperglycemia (70–90) - Cholelithiasis (20–35)	No data available in mHC Hypothetical use in pituitary mHC in the absence of diabetes
Cabergoline	DA receptors - DR2: corticotrophs inhibition - DR1: adrenal cortex cells stimulation	- 0.5–0.7 mg PO every wk - CD	Never tested	- Nausea (rare) - Depression (rare)	No data available in mHC The effect more likely in patients with less severe hypercortisolism
Metyrapone	Steroidogenesis inhibitor - 11-βhydroxylase inhibition - 18-hydroxylase inhibition (lesser degree) Tissue GC metabolism - 11BHSD inhibition Pituitary - Inhibition of GC-driven positive-feedback (subtypes corticotropinomas) - SS and DA receptors expression (β-arrestin 1 and β-arrestin 2)	- 250–1500 mg PO qid - CS of any origin	250–500 mg bid (late afternoon and evening)	- Female hirsutism (36) - Dizziness (30) - Arthralgias (15)	Correction of the abnormal circadian rhythm of cortisol if given in the late afternoon and evening Presurgery short-term
Ketoconazole	Steroidogenesis inhibitor - 11-βhydroxylase inhibition - 18-hydroxylase inhibition - 17a-hydroxylase inhibition - 20,22-desmolase inhibition - 17,20-desmolase inhibition Pituitary - Inhibition of corticotrophs (adenylate cyclase)	- 200–800 mg PO bid-tid - CS of any origin	200–400 mg/day	- Epatotoxicity (14.5) - Nausea (12.9) - AI (11.9) - Gynecomastia (17)	Case report showing cortisol secretion normalization and blood pressure amelioration with low dose (200–400 mg/day)
Levoketoconazole	Steroidogenesis inhibitor - 11-βhydroxylase inhibition - 18-hydroxylase inhibition - 17a-hydroxylase inhibition - 20,22-desmolase inhibition - 17,20-desmolase inhibition Pituitary - Inhibition of corticotrophs (adenylate cyclase)	- 150–600 mg PO bid-qid - CS of any origin	Never tested	- Nausea (30) - Edema (19) - Headache (28)	No data available in mHC In patients with type 2 diabetes mellitus without hypercortisolism amelioration of glycometabolic control, lipid levels, body weight and blood pressure
Osilodrostat	Steroidogenesis inhibitor - 11-βhydroxylase inhibition	- 5–60 mg PO qd-bid - CS of any origin	Never tested	- AI symptoms (32–52) - Asthenia (30–58) - Nausea (32–42)	No data available in mHC
Mifepristone	Non selective GC receptor antagonist	- 300–1200 mg PO - CS of any origin	200–400 mg	- Nausea (48) - Asthenia (48) - Headache (44)	Three studies with amelioration of insulin resistance, hypertension, QoL and cardiometabolic parameters, with good tolerability
Relacorilant	Selective GC receptor antagonist	- 100–400 mg PO qd - CS of any origin	Never tested	- Back pain (31) - Headache (26) - Edema (26)	A phase III, randomized, double-blind, placebo-controlled study (NCT04308590) ongoing

Footnotes: CS: Cushing’s syndrome; mHC: Mild hypercortisolism; PO (per os): Orally; IM: intra-muscularly; wk: Week; SC: Subcutaneously; qd (quaque die): Once daily; bid (bis in die): Twice daily; tid (ter in die): 3 times a day; qid (quater in die): 4 times a day; DA: Dopamine; CD: Cushing’s disease (pituitary hypercortisolism); DR2: DA receptor type 2; DR1: DA receptor type 1. SS: Somatostatin; 11BHSD1: 11 beta-hydroxysteroid dehydrogenase type 1; GC: Glucocorticoid; AI: Adrenal insufficiency. QoL: Quality of life. Cabergoline could exert a bimodal effect in patients with pituitary mHC, as it may inhibit ACTH secretion, by acting on D2R at pituitary level, while it may induce the steroidogenesis, by acting on D1R at the adrenal level, this latter mechanism being possibly responsible of the drug escapes. Metyrapone, in addition to directly inhibiting steroidogenesis, may reduce ACTH secretion in those corticotropinomas that are responsive to the GC-driven positive feedback and it could reduce ACTH secretion by modulating the SS and DA receptors expression thanks to its role on β-arrestin 1 and β-arrestin 2.

## Data Availability

Not applicable.

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
