# Peer review of "Management and Medical Therapy of Mild Hypercortisolism"

_ijms, 2021, doi:10.3390/ijms222111521_

Round 1

Reviewer 1 Report

Reviewer #1: In this manuscript, the researchers tried to explain about the Management and medical therapy of mild hypercortisolism. It is interesting work and can be accepted after revision.

-     The grammar errors should be checked in the whole manuscript.

-       In abstract, the first four lines should be summarized.

-       In introduction, the main objective has been repeated so it should be refined.

-       Some recent and relevant articles may be added as thousands of articles have been published on this topic.

-       Conclusion should be refined as it is not properly written as per results.

Author Response

Reviewer #1: In this manuscript, the researchers tried to explain about the Management and medical therapy of mild hypercortisolism. It is interesting work and can be accepted after revision.

- The grammar errors should be checked in the whole manuscript.

We apologize. The grammar errors have been checked and corrected throughout the manuscript

-  In abstract, the first four lines should be summarized

As suggested, in the Abstract the first two sentences have been summarized (lines 16-18).

-  In introduction, the main objective has been repeated so it should be refined.

We thank the reviewer for the suggestion. The Introduction has been modified as suggested (lines 53-61, 97-98). In particular, the difference between the HidHyCo condition and the mHC condition has been clarified

-  Some recent and relevant articles may be added as thousands of articles have been published on this topic.

We agree. We have performed a further search and added the most significant studies that had not been previously included (lines 171-173, 253-260, 307-309, 380-382, 477-479, 505, 510-512, 544-546). Although the present review was focused on the phase 3 studies, we have also added some very recent data on possible agents that could become available in the next future

-  Conclusion should be refined as it is not properly written as per results

The conclusion have been rewritten in accordance with the reviewer suggestion (lines 623-627, 634-638)

Reviewer 2 Report

The review by Favero et al. summarizes the available data regarding the medical treatment of mild hypercortisolism (mHC) focusing on the molecular pathways that are targeted by the available drugs. The manuscript is well organized and detailed.

Author Response

The review by Favero et al. summarizes the available data regarding the medical treatment of mild hypercortisolism (mHC) focusing on the molecular pathways that are targeted by the available drugs. The manuscript is well organized and detailed.

We thank the reviewer for the comment